# Bacterial AB$_5$ toxins inhibit the growth of gut bacteria by targeting ganglioside-like glycoconjugates

Robert T. Patry[1,2], Martin Stahl[3], Maria Elisa Perez-Munoz[4], Harald Nothaft[2], Cory Q. Wenzel[2], Jessica C. Sacher[2], Colin Coros[5], Jens Walter[2,4], Bruce A. Vallance[3] & Christine M. Szymanski[1,2]

The AB$_5$ toxins cholera toxin (CT) from *Vibrio cholerae* and heat-labile enterotoxin (LT) from enterotoxigenic *Escherichia coli* are notorious for their roles in diarrheal disease, but their effect on other intestinal bacteria remains unexplored. Another foodborne pathogen, *Campylobacter jejuni*, can mimic the GM1 ganglioside receptor of CT and LT. Here we demonstrate that the toxin B-subunits (CTB and LTB) inhibit *C. jejuni* growth by binding to GM1-mimicking lipooligosaccharides and increasing permeability of the cell membrane. Furthermore, incubation of CTB or LTB with a *C. jejuni* isolate capable of altering its lipooligosaccharide structure selects for variants lacking the GM1 mimic. Examining the chicken GI tract with immunofluorescence microscopy demonstrates that GM1 reactive structures are abundant on epithelial cells and commensal bacteria, further emphasizing the relevance of this mimicry. Exposure of chickens to CTB or LTB causes shifts in the gut microbial composition, providing evidence for new toxin functions in bacterial gut competition.

[1] Department of Microbiology and Complex Carbohydrate Research Center, University of Georgia, Athens, GA 30602, USA. [2] Department of Biological Sciences, University of Alberta, Edmonton, AB T6G 2E9, Canada. [3] Division of Gastroenterology, BC Children's Hospital Research Institute, The University of British Columbia, Vancouver, BC V6H 3V4, Canada. [4] Department of Agricultural, Food and Nutritional Science, University of Alberta, Edmonton, AB T6G 2E1, Canada. [5] Delta Genomics, Edmonton, AB T5J 4P6, Canada. Correspondence and requests for materials should be addressed to C.M.S. (email: cszymans@uga.edu)

The gut microbiome is an exceedingly complex ecosystem where bacteria employ every tactic at their disposal to gain an advantage over competitors while avoiding clearance by the host. Among these strategies, bacteria display a diverse array of surface glycan structures[1] and mimic host glycans[2–5] to evade immune recognition. Given that these structures are prominently exposed, many animals have developed innate glycan-binding proteins, such as toll-like receptors or siglecs[6], to target non-self pathogen-associated molecular patterns and aid in immune clearance. There are also several glycan binding proteins such as galectins[7], intelectins[8] and resistin-like molecule β (RELMβ)[9] that directly inhibit bacterial spread in the intestine. This study explores the ability of bacteria to use glycan binding proteins to target each other, contributing to competition between groups within the intestinal ecosystem.

Cholera toxin (CT) is produced by *Vibrio cholerae*, the causative agent of cholera, a devastating gastrointestinal disease currently endemic in 51 countries[10]. The toxin is a member of the AB$_5$ group of bacterial toxins and consists of 5 B-subunits that specifically target cell surface glycan receptors and trigger entry into the host[11]. AB$_5$ toxins also contain an enzymatic A-subunit responsible for the toxic effect observed. The A-subunit of CT ADP-ribosylates the G$_{s\alpha}$ protein of adenylate cyclase, causing it to bind GTP and constitutively stimulate conversion of ATP to cyclic AMP (cAMP)[12]. This leads to increased efflux of ions into the intestinal lumen, resulting in the rice-water diarrhea and severe dehydration that are hallmarks of the disease[13]. The B-subunits of CT (CTB) are arranged in a symmetrical, ring-shaped pentamer[14,15]. It was originally believed that CTB bound to GM1 gangliosides on the apical surface of intestinal epithelial cells to achieve entry[16]. This hypothesis predominated in part due to CT's exceptionally high affinity for its receptor, with a reported dissociation constant of 0.73 nM[17]. However, the dogma that binding to the intestinal epithelium is mediated through GM1 gangliosides has recently been disputed due to observations that these receptors are sparsely present in the human GI tract[18,19]. Further reports demonstrate that CT is capable of binding to fucosylated structures that serve as functional receptors for the toxin[19–21].

Heat-labile enterotoxin (LT) from enterotoxigenic *Escherichia coli* (ETEC) is another AB$_5$ toxin that shares a similar structure and function to CT. Both exhibit nM affinities for GM1 ganglioside receptors, though recent studies have identified other, albeit lower-affinity, receptors for these toxins. LT is capable of binding to blood group antigens, and both CT and LT were shown to bind several human milk oligosaccharides, notably those containing fucosylated residues[20]. It has been shown that the fucose binding sites on these toxins are distinct from those binding GM1 gangliosides[20], leading us to question the role of the high affinity GM1 binding sites on these toxins. Notably, CT has been used as a reagent to assess GM1 ganglioside mimicry of another gastrointestinal pathogen, *Campylobacter jejuni*[22].

*C. jejuni* is a leading cause of gastroenteritis worldwide[23,24]. In addition, *C. jejuni* can cause the serious post-infectious sequelae Guillain-Barré Syndrome (GBS) through its ability to mimic human gangliosides with the lipooligosaccharide (LOS) structures displayed on its surface[25]. Gangliosides are glycolipids containing sialic acid in their carbohydrate structure and are attached to ceramide lipids. These structures commonly decorate nerve cells but can also be found on other cells throughout the body. LOS is a shorter lipopolysaccharide (LPS) lacking the O-antigen and consists of a lipid A portion, which anchors it to the cell wall, as well as inner and outer core oligosaccharides. Among *C. jejuni* strains, the outer core structure exhibits considerable variation in the type and arrangement of the saccharides presented. Many *C. jejuni* strains sialylate their LOS, enabling them to mimic human gangliosides[25].

Approximately 60% of *C. jejuni* strains can mimic gangliosides[26], including GM1, GM2, GM3, GD1a, GD1b, GD2, GD3, and GT1a ganglioside types[27]. This subset encompasses most of the major gangliosides found in the human body, and mimicry of these structures is believed to allow *C. jejuni* to escape immune detection by their hosts. GBS occurs when there is a breakdown in immune tolerance and the host generates α-ganglioside antibodies that not only attack the pathogen but subsequently recognize host nerve cells as foreign. This leads to degradation of spinal nerve axons and paralysis[28]. GM1 gangliosides are the most common gangliosides mimicked by *C. jejuni* glycans (structures depicted in Fig. 1), and CT has been used to probe for the expression of these structures on its surface[22].

The ability of CTB to bind *C. jejuni* is strain-dependent, due to variation in LOS structures resulting from differences in biosynthetic enzymes and/or variation in the expression of terminal sugar transferases[27]. The heat-stable (HS) serostrain HS:19, along with its serogroup members, have most commonly been isolated from GBS patients and express a heterogeneous LOS outer core displaying a mixture of GM1 and GD1a mimics[29]. Conversely,

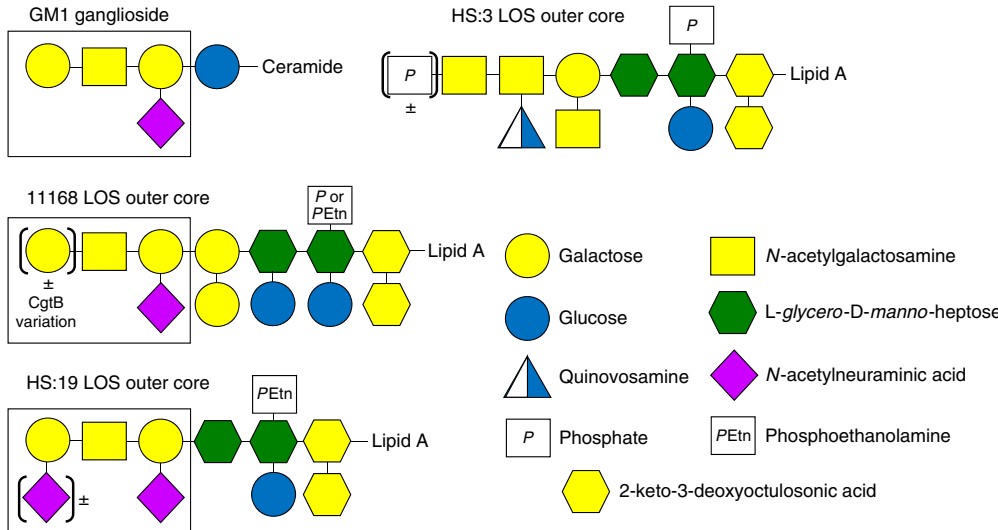

**Fig. 1** Structures of GM1 ganglioside and the outer LOS core of the wildtype *C. jejuni* strains used in this study. The portion of each receptor that is recognized by α-GM1 antibodies and cholera toxin is indicated by the box

the HS:3 serostrain is incapable of mimicking gangliosides since the strain lacks enzymes to synthesize the appropriate sugars[26]. The 11168 strain of *C. jejuni* is capable of phase-varying its *cgtB* gene (*cj1139c*), which encodes the β 1–3 galactosyltransferase responsible for addition of a terminal galactose onto the LOS[22]. This strain can vary its GM1 ganglioside mimics through slipped-strand mispairing in the poly-guanosine tract within *cgtB* (Fig. 1) resulting in premature truncation of the transferase and loss of mimicry[22].

In this study, we demonstrate that CTB and LT increase cell membrane permeability and inhibit the growth of GM1-mimicking *C. jejuni* strains, presenting novel functions for these toxins. This effect is observed void of the toxic A subunit of each toxin and selects for *C. jejuni* variants not expressing GM1 mimicry. In addition, we identify another CT-binding bacterium in the chicken gut, *Enterococcus gallinarum/casseliflavus*, and provide evidence that oral administration of CTB and LT B-subunit (LTB) to chickens induces changes in the gut microbiome.

## Results

**Cholera toxin shows GM1-dependent binding to *C. jejuni*.** A schematic of the LOS outer core structures of each strain used in this study is depicted in Fig. 1, along with GM1 ganglioside for comparison. CTB binding to *C. jejuni* LOS was compared by transmission electron microscopy and immunogold labeling (Fig. 2a–c). CTB binding was dependent on the presence of LOS-mimicking GM1 structures. *C. jejuni* HS:19 cells, which express GM1 mimics[29], showed toxin binding (Fig. 2a), whereas *C. jejuni* HS:3 cells unable to mimic GM1, showed no toxin binding (Fig. 2c). *C. jejuni* 11168, which is known to display a mixed population of LOS variants[22], showed a mixed CTB binding phenotype (Fig. 2b).

**The B-subunits of CT and LT clear *C. jejuni* growth.** CT, CTA, CTB and LTB were spotted on agar plates containing different *C. jejuni* strains to determine the effect of toxin binding on the growth of each strain. A zone of clearance was observed when CT, CTB, or LTB were spotted on *C. jejuni* HS:19 (Fig. 2d). This phenotype was not observed for strain HS:3, which does not bind CT (Fig. 2f), while the partial clearance observed for strain 11168 reflects the mixed binding phenotype (Fig. 2e). The clearance caused by CTB suggests that the enzymatically active A-subunit known to be toxic to eukaryotic cells was not necessary to cause the clearance phenotype. LTB, the B-subunit of the related AB₅ toxin also showed similar clearance to that seen with CTB (Fig. 2). When these same treatments were applied to *E. coli* CWG308 pGM1a/pCst, a strain engineered to display the GM1 ganglioside mimic on its LOS, no clearance was observed (Supplementary Fig. 1).

To confirm that the observed clearance was mediated through the GM1 ganglioside-binding domain of the toxin and not through the fucose-binding domain, CTB was mixed at a 1:1 ratio with GM1 ganglioside, GM2 ganglioside or fucose prior to spotting onto *C. jejuni* HS:19-containing plates (Fig. 2g). The GM1 ganglioside competitively inhibited the clearance effect of CTB, while neither pre-incubation with GM2 or with fucose had an effect, indicating that the GM1 ganglioside-binding domain of the toxin is required for the clearing activity. The minimum amount of toxin required to see this effect was determined to be 0.25 μg (4.3 μM) by spotting various concentrations of CTB onto *C. jejuni* HS:19 and observing the extent of clearance in each zone (Fig. 2h).

**CTB mediated clearance is bacteriostatic.** To better understand the clearance phenotype, scanning electron microscopy was used to examine the interface between the clearance/growth zone on *C. jejuni* HS:19 plates. Squares of agar were excised from either

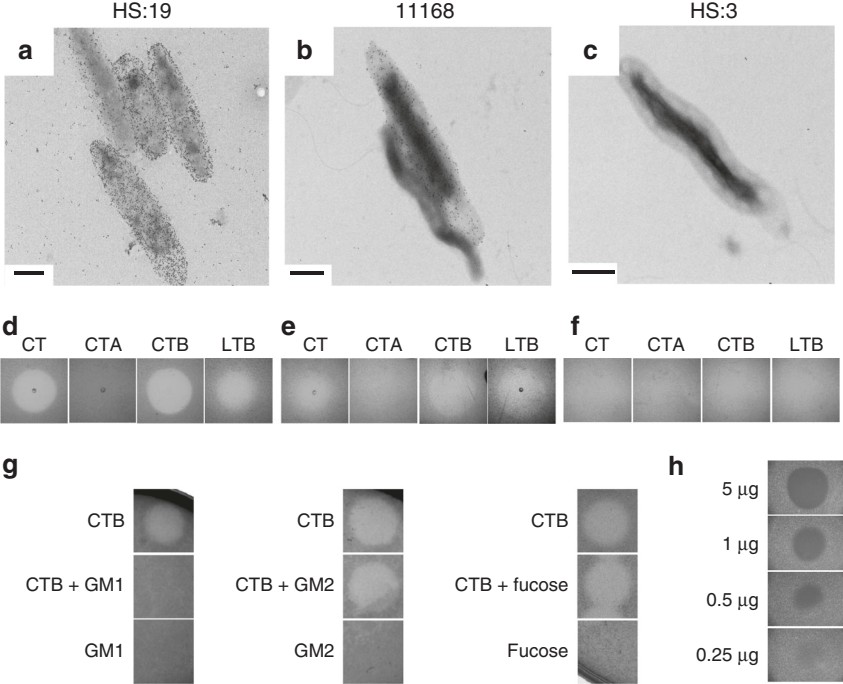

**Fig. 2** Cholera toxin B subunit (CTB) and heat-labile enterotoxin B subunit (LTB) bind and clear GM1 ganglioside-mimicking *C. jejuni* strains. **a–f** Transmission electron micrographs depicting different *C. jejuni* strains bound by immunogold-labeled CTB (**a–c**, scale bars are 0.5 μm), accompanied by images of agar plates showing *C. jejuni* following exposure to 2 μL of either CT, CTB, CTA, or LTB (each at 1 mg mL⁻¹) for 24 h while growing in soft agar (**d–f**). **a, d** *C. jejuni* HS:19. **b, e** *C. jejuni* 11168. C, F) *C. jejuni* HS:3. **g** Competitive clearance assay where CTB was mixed 1:1 with indicated glycans prior to spotting onto growing *C. jejuni* HS:19. **h** CTB spotted on *C. jejuni* HS:19 in various concentrations to determine the amount needed for clearance

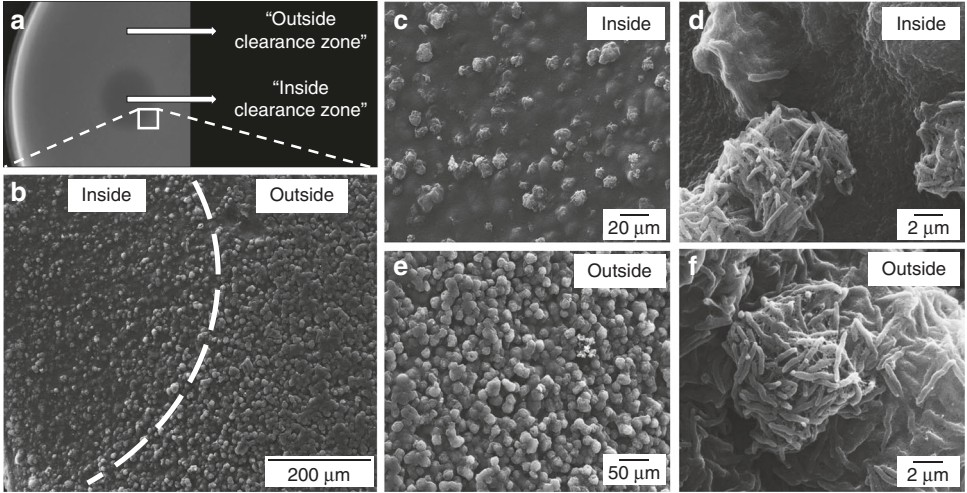

**Fig. 3** Scanning electron micrographs depicting *C. jejuni* HS:19 grown in NZCYM soft agar following exposure to cholera toxin B subunit (CTB). **a** Clearance zone 24 h after spotting CTB (the white square indicates the location of the excised agar slab). **b** A low magnification image highlighting the difference in surface structure of the agar slab between the exposed area (left side) and the unexposed area (right side). **c, d** Increasingly magnified images of the agar surface inside the zone of clearance. **e, f** Increasingly magnified images of the agar surface outside the zone of clearance. Images are representative of 3 replicate experiments

inside, outside, or directly on the edge of the clearance zones, to visualize the growth simultaneously (Fig. 3a). When viewed at low magnification, regions outside of the CTB clearance zone showed growth in many discrete spheres, whereas these spheres were more sparsely distributed within the clearance zone (Fig. 3b). This difference was also apparent at higher magnifications, where a drastic decrease in the number of *C. jejuni* cells inside the clearance zone was observed compared to outside (Fig. 3c vs. 3e), but all showed normal cell morphologies (Fig. 3d, f, Supplementary Fig. 2A and 2B). Addition of CTB to agar plates with already existing *C. jejuni* growth did not cause clearing (Supplementary Fig. 2D). These results support the hypothesis that toxin exposure leads to decreased *C. jejuni* growth in the observed zones of clearance.

**Changes in LOS and *cgtB* suggest selection against mimic**. To analyze whether cells exposed to CTB and LTB can alter their LOS structures and phase-vary GM1 ganglioside mimics, the LOS was isolated from *C. jejuni* 11168 cells inside and outside CTB and LTB clearance zones. LOS from each was compared using SDS-PAGE followed by silver staining to visualize molecular weight differences in the LOS (Fig. 4a), and then by far western blotting with CTB to verify decreased LOS recognition by CTB (Fig. 4b). The silver stained LOS from cells isolated from outside the zones showed a doublet (indicated by the arrow) and was reduced to a single lower molecular weight band for LOS isolated from cells inside the clearance zones (Fig. 4a). The top band of the doublet (Fig. 4a, lane 1) was presumed to represent the full length LOS and the lower band to be lacking the terminal galactose residue. The LOS single band from cells inside the zones migrates similar to the lower of the two bands of the doublet from outside the clearance zone. To confirm this, a far western blot probed with CTB was done on the same samples and showed much greater CTB binding to LOS isolated from cells unexposed to CTB/LTB compared with those that had already been exposed (Fig. 4b). The shift in mass and reduction in CTB binding observed in CTB/LTB-exposed cells suggests that exposure to the toxin provides a selective pressure against GM1 ganglioside mimicry in *C. jejuni*.

To determine whether sequence variation in *cgtB* (necessary for addition of the terminal galactose in generating the GM1 mimic)

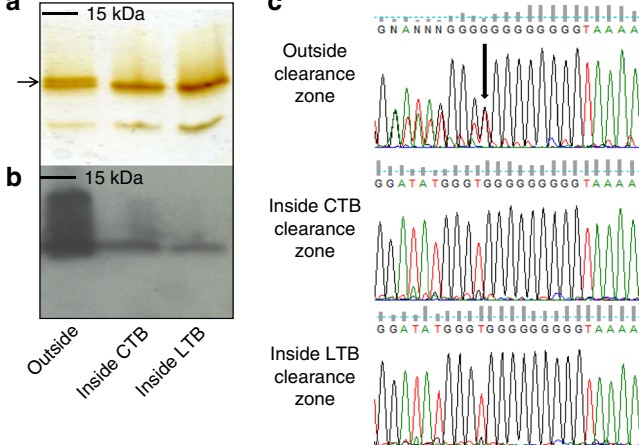

**Fig. 4** Exposure of *C. jejuni* 11168 cells to cholera toxin B subunit (CTB) led to changes in lipooligosaccharide (LOS) structure. This is shown by silver stain (**a**) and far western blot using CTB as a probe (**b**). CTB exposure also led to changes in the length of the poly-G tract in the *cgtB* gene, which is essential for GM1 ganglioside mimicry, leading to a frameshift mutation (**c**). The arrow in **a** marks the two bands associated with two distinct LOS structures in wildtype 11168 and the arrow in **c** marks the ninth G in the homopolymeric tract where heterogeneity is seen in the sequence. Source data are provided as a Source Data file

led to the alteration in *C. jejuni* LOS following CTB or LTB exposure, DNA was isolated from *C. jejuni* 11168 cells inside and outside the same clearance zones. The *cgtB* gene was then PCR-amplified and sequenced from each sample. Interestingly, cells outside the zone of clearance displayed sequence heterogeneity at the homopolymeric G tract in *cgtB*, indicative of a population comprised of cells containing either 8 or 9 Gs in this region (Fig. 4c). However, this heterogeneity was not present for cells obtained inside the zones of clearance, which consistently showed a sequence of 9 Gs. This shift from 8 to 9 Gs causes a frameshift during *cgtB* mRNA translation, which creates a premature stop codon resulting in a non-functional protein and abrogation of GM1 ganglioside mimicry[22]. It is apparent from our data that

**Table 1 Strains and mutants used in this study**

| Strain or mutant | Description | Source |
|---|---|---|
| *C. jejuni* HS:19 serostrain | Human clinical isolate displaying both GM1 and GD1a ganglioside mimics. | ATCC 43446[67] |
| *C. jejuni* HS:3 serostrain | Isolate incapable of mimicking mammalian gangliosides due to lack of sialic acid biosynthesis genes. | ATCC 43431[67] |
| *C. jejuni* 11168 | Human clinical isolate displaying GM1 and GM2 gangliosides dependent on the phase-variable state of the *cgtB* gene. | 68 |
| *C. jejuni* 11168 Δ*cgtB::kan^R* | Kanamycin cassette knockout mutant in the *cgtB* gene incapable of adding terminal galactose to generate GM1 ganglioside mimic. | This study |
| *C. jejuni* 11168 Δ*cmeA::cm^R* | Chloramphenicol cassette knockout mutant in the *cmeA* gene creating a non-functional multi-drug efflux pump. | This study |
| *C. jejuni* 11168 Δ*neuC1::kan^R* | Kanamycin cassette knockout mutant in the *neuC1* gene incapable of creating N-acetylneuraminic acid to generate GM1 ganglioside mimic. | This study |
| *E.coli* CWG308 | *waaO* mutant of *E. coli* R1 strain F470 with truncated LPS containing only the lipid A and inner core portions. | 69 |
| *E.coli* CWG308 pGM1a/pCst | A mutant made in *E. coli* CWG308 which harbors *C. jejuni cgtA, cgtB* and *cstII,* as well as *Neisseria lgtE* and *E. coli gne* on two separate plasmids, to display LOS GM1 ganglioside mimics. | 70 |
| *Enterococcus gallinarum/ casseliflavus* | A GM1 ganglioside-mimicking bacterium isolated from the chicken cecum. | This study |

*C. jejuni* 11168 cells inside CTB- and LTB-induced zones of clearance have "switched off" expression of *cgtB*, supporting the hypothesis that exposure to CTB or LTB constitutes a selective pressure against GM1 mimic expression.

**Exposure to CTB increases cell permeability to EtBr.** To determine if the inhibitory effect of CTB binding was due to a change in membrane permeability, accumulation of ethidium bromide (EtBr) inside the cells was measured by spectrophotometry. EtBr has been used previously to measure cell membrane permeability changes, as well as activity of multi-drug efflux pumps[30,31]. *C. jejuni* 11168 was used for this experiment along with several isogenic mutants (Table 1) including two abolishing GM1 mimicry (*cgtB* and *neuC1*) and one in the major efflux pump gene, *cmeA*. As seen in Fig. 4, addition of CTB to the *C. jejuni* wildtype, capable of mimicking GM1 gangliosides, resulted in an increased influx of EtBr unrelated to *C. jejuni* efflux pump activity. The relative fluorescence recorded 20 min after EtBr addition changed significantly when 11168 wildtype cells had been exposed to CTB ($n = 21$, dF $= 40$, $t = 4.3294$, $p = 9.74 \times 10^{-5}$, $t$-test) and as well for Δ*cmeA* ($n = 12$, dF $= 22$, $t = 2.4580$, $p = 0.0223$, $t$-test); however no difference was observed when the toxin could no longer bind in Δ*cgtB* ($n = 12$, dF $= 22$, $t = 0.9388$, $p = 0.3580$, $t$-test) and Δ*neuC1* ($n = 12$, dF $= 22$, $t = 1.4302$, $p = 0.1667$, $t$-test). Consistent with the spot assay, we did not see a change in EtBr uptake following CTB treatment of *E. coli* CWG308 pGM1a/pCst ($n = 6$, dF $= 10$, $t = 1.8869$, $p = 0.0885$, $t$-test) (Supplementary Fig. 1D). *E. coli* CWG308 wildtype showed a statistically significant increase in fluorescence after CTB exposure ($n = 9$, dF $= 16$, $t = 3.8268$, $p = 0.0015$, $t$-test), but this difference was small in comparison to that seen for *C. jejuni* 11168 and the significance may result more from low data variance (Supplementary Fig. 1D). Relative fluorescence measurements were taken every 2 min, but the trends remained relatively consistent throughout (Supplementary Fig. 3). These findings support that CTB causes an increase in membrane permeability upon binding to *C. jejuni* and is correlated with strains that show the clearance phenotype.

**CTB reduces *C. jejuni* virulence in *Galleria mellonella* model.** The *G. mellonella* model is commonly used to assess *C. jejuni* virulence due to its susceptibility to *C. jejuni* infection and mortality phenotype in response to virulent strains[32]. To test the potential impact of CTB on *C. jejuni* pathogenicity, ~ $1.0 \times 10^9$ colony forming units of *C. jejuni* HS:19 was introduced into the hind-leg of *G. melonella* to assess its virulence in the presence of CTB. When *C. jejuni* was incubated with CTB prior to injection, a significant increase ($n = 9$, dF $= 16$, $t = 4.3374$, $p = 0.0005$, $t$-test) in larvae survival was observed after 5 days infection compared to HS:19 administered alone (Fig. 5). This indicates that CTB binding to *C. jejuni* has a significant impact on its ability to infect and cause disease in a model host.

**CTB binds to mucins, epithelium and commensal bacteria.** Chickens are a common host for *C. jejuni* so cloacal swabs were performed on 10% of chickens upon arrival to confirm that we received a Campylobacter-negative flock by plating onto Karmali selective agar. At the termination of the experiment, cecal contents from all birds were serially diluted and plated onto Karmali agar to confirm the chickens were not colonized with *C. jejuni*. To determine whether chicken intestinal epithelial cells or chicken gut bacteria other than *C. jejuni* display GM1 mimics or fucosylated structures capable of CTB binding, chicken ceca and their contents were fixed, and cross-sections were paraffin-embedded and sectioned for analysis by immunofluorescent microscopy. Paraffin cross-sections were probed using either CTB or α-GM1 ganglioside antibodies. CTB bound strongly to the apical surface of the epithelium, to the mucus layer and to mucin-filled goblet cells (Fig. 6a, c). A majority of the binding to mucus was negated by prior treatment of the slides with α1-2-fucosidase (Fig. 6d), suggesting that this binding was mediated by the fucose binding site of CTB. CTB also bound to commensal bacteria present within the cecal lumen (Fig. 6e). Similar to CTB, α-GM1 ganglioside antibodies labeled both the apical border of the epithelium, as well as many commensal bacteria within the cecal lumen (Fig. 6b, f). However, unlike CTB, the α-GM1 ganglioside antibody did not bind the mucus or goblet cells (Fig. 6a, b). This indicates that CTB may bind to the apical epithelium and to bacteria colonizing the chicken intestine via its GM1-binding site, since it shares specificity for these structures with αGM1 antibodies. In addition, our evidence suggests that CTB binds mucus and goblet cells via its fucose receptor, as binding to these components was abrogated by fucosidase treatment and since specificity for these components is not shared with α-GM1 antibodies.

**LTB administration to chickens shifts intestinal microbiota.** Given the evidence that AB₅ toxins bind to a subset of bacteria

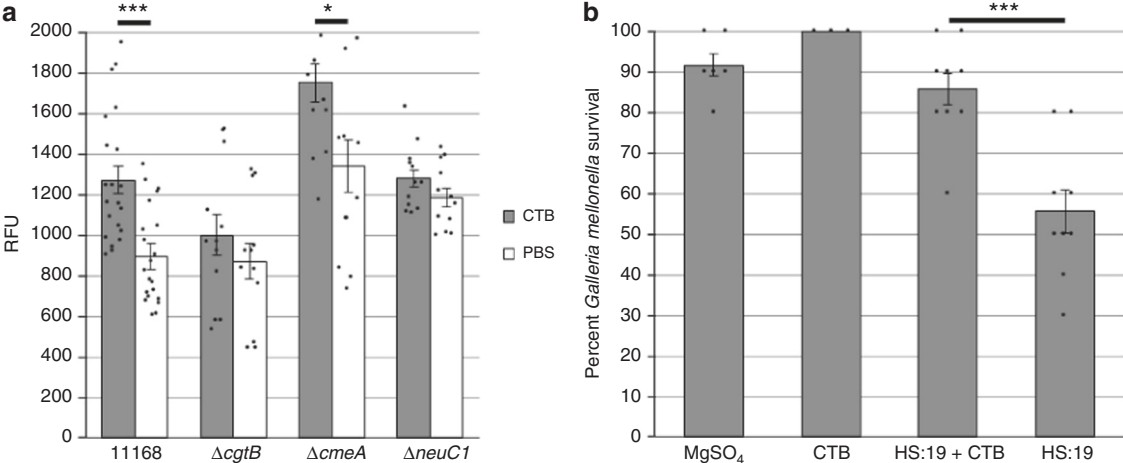

**Fig. 5** Cholera toxin B subunit (CTB) increases membrane permeability and reduces *C. jejuni* virulence in the *Galleria mellonella* model. **a** *C. jejuni* 11168 wildtype ($n = 21$ per condition), $\Delta cgtB$ ($n = 12$ per condition), $\Delta cmeA$ ($n = 12$ per condition) and $\Delta neuC1$ ($n = 12$ per condition) were incubated with ethidium bromide for 20 min following treatment with PBS (white) or CTB (gray) and the relative fluorescence when exposed to CTB was measured. **b** *G. mellonella* larvae were inoculated with *C. jejuni* HS:19 alone ($n = 9$) or together with CTB ($n = 9$), MgSO₄ buffer alone ($n = 6$), or MgSO₄ buffer together with CTB ($n = 3$), and larvae survival rates were determined after 5 days. Error bars represent the standard error within each group. ***p-value ≤ 0.001, *p-value ≤ 0.05 determined by two-tailed *t*-test. Source data are provided as a Source Data file

present in the chicken intestine, we next examined if such toxins impact the composition of the chicken intestinal microbiota. We compared the 16S rRNA gene sequence profile of the cecal bacterial communities in chickens treated with LTB ($n = 9$) or with PBS ($n = 10$) as a control (dF = 17). We focused this experiment on LTB since enterotoxin producing *E. coli* strains have been reported in chickens[33,34], and producing these toxins may be ecologically relevant. Administration of LTB causes shifts in the overall composition of the gut microbiome relative to PBS-treated chickens as shown by NMDS analysis of Bray Curtis dissimilarity ($p = 0.0120$ by Adonis PERMANOVA test) (Fig. 7a). Although α-diversity is not affected, LTB significantly reduces β-diversity of the cecal bacterial community (Fig. 7b), indicating that the toxin lowers inter-individuality of the gut microbiota. In terms of specific taxa, we found that LTB caused major shifts in the abundance of bacteria classified within the phyla Firmicutes ($p = 0.04$) and Bacteroidetes ($p = 0.04$) (Fig. 7c, Table 2, statistical significance determined by two-tailed unpaired *t*-test with Welch's correction). Analysis of relative abundance at lower taxonomic levels shows that the majority of changes involve decreases in the families Ruminococcaceae ($p = 0.02$) and Lachnospiraceae ($p = 0.05$), as well as genera and OTUs within these families, while the genus *Bacteroides* was increased ($p = 0.04$) (Fig. 7d, Table 2). These findings demonstrate that LTB has a major effect on the chicken gut microbiota.

To test if similar findings apply to CTB, we sequenced the gut microbiota of three birds receiving CTB and three with PBS. Although statistical significance could not be obtained, we observed similar trends to those obtained with LTB (Supplementary Fig. 4). Overall, our findings indicate that AB₅ toxins affect more than just *C. jejuni* in the chicken gut.

**Isolation and identification of GM1-mimicking Firmicutes.** Upon finding evidence to support other GM1-ganglioside mimicking bacteria in the chicken gut impacted by AB₅ toxins, α-GM1-protein G agarose was used to pull-down bacteria that display ganglioside mimics on their surface. Isolated bacteria were screened and one GM1-positive species, confirmed to bind α-GM1 and CTB by fluorescence microscopy, was further characterized. The isolate was identified as *Enterococcus gallinarum/casseliflavus*, a member of the Firmicutes phylum. Fluorescent microscopy of

the isolated strain with α-GM1 ganglioside antibody showed two distinct populations with only a fraction of cells being bound by the GM1 antibody (Fig. 6g), Further pull-downs with CTB in combination with α-CTB and protein G agarose intended to isolate the GM1-ganglioside mimicking variants (Fig. 6h) followed by microscopic analyses after CTB-α-CTB surface staining resulted in a similar phenotype with only a fraction of bacteria being positive for CTB binding. This indicates that only a fraction of the culture expresses the potential GM1 mimics *in vivo*, reminiscent of potential phase-variation as observed in *C. jejuni* 11168.

**Discussion**
It has long been considered that the primary role of bacterial toxins is to target vertebrate hosts and cause disease[35]. Another assumption, that has recently been disputed, is that GM1 gangliosides in the human gut are necessary for CT to induce disease in humans[19]. However, since CT is capable of binding to *C. jejuni* GM1 ganglioside mimics[22], and the toxin genes are encoded by a *V. cholerae* bacteriophage[36], we thought it was possible for CT to possess an antibacterial function. *C. jejuni* is capable of mimicking various host gangliosides through variation of its LOS outer core, which allows the organisms to evade host immune detection[37,38]. Furthermore, *C. jejuni* is an intestinal pathogen that occupies a niche shared with other enteric pathogens such as *V. cholerae*, suggesting the two organisms could compete for resources[37].

In endemic areas, *C. jejuni* is rarely the sole enteric pathogen infecting an individual, and the organism is routinely isolated from humans co-infected with *V. cholerae* and/or enterotoxigenic *E. coli*[39,40]. The latter produces LT, another AB₅ toxin with binding and enzymatic properties remarkably similar to CT[41]. AB₅ toxin-producing bacteria have also been found in chickens, a common commensal host for *C. jejuni*[33,34,42–44]. As a result, *C. jejuni* may come into contact with ganglioside-targeting, AB₅ toxin-producing bacteria in its natural environment. This study has demonstrated that CT and LT do in fact inhibit growth of GM1-mimicking *C. jejuni*. In addition, we have shown that when fed to chickens, LTB (and CTB) alter the microbial composition of the chicken intestinal tract, suggesting that these toxins and others like it may indeed play a role in inter-bacterial warfare.

After confirming that CT binding to *C. jejuni* was dependent on GM1 ganglioside mimicry and that this binding inhibited

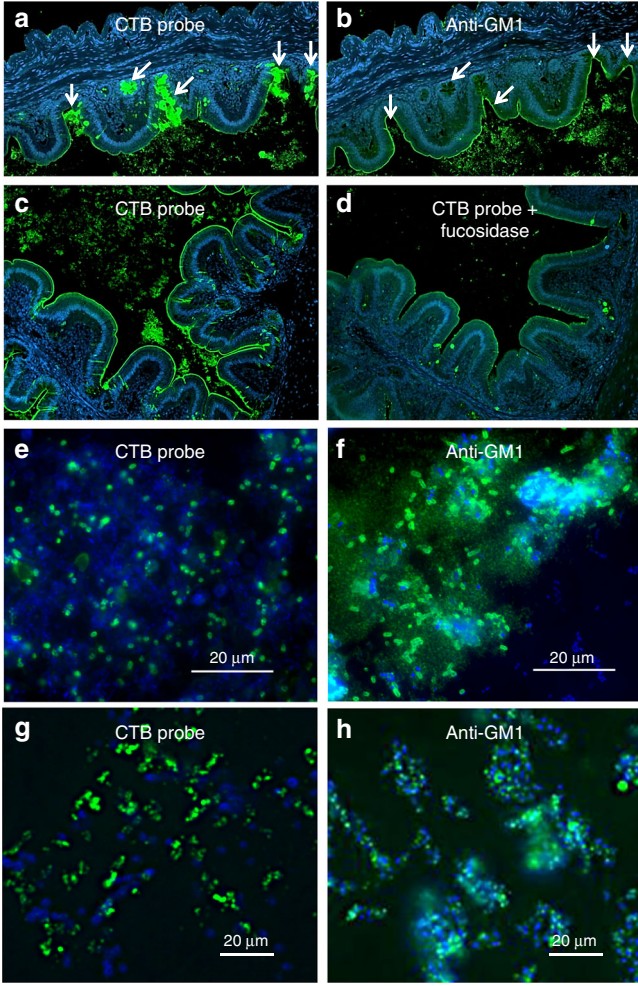

**Fig. 6** Cholera toxin (CT) binds to goblet cells, mucins and commensal bacteria in the chicken cecum. Fluorescence microscopy of sequentially cut chicken cecal cross-sections labeled with DAPI (blue) and either α-CT (green) (**a**, **c**–**e**, **g**) or α-GM1 ganglioside (green) (**b**, **f**, **h**). Images **a**–**d** were taken at ×200. Arrows in **a**, **b** indicate goblet cells and mucus. **d** Cells were treated with fucosidase prior to labeling. **e**, **f** Magnified images to show bacterial binding. **g** CTB (green) and DAPI (blue) labeling of bacteria isolated after CTB-protein G pulldown of bacteria isolated from the initial GM1-protein G pull-down **h** Bacteria isolated after GM1 pulldown and labeled with DAPI (blue) and α-GM1 (green)

*C. jejuni* growth, we tested which component of the $AB_5$ toxins mediated the antibacterial activity. Interestingly, we found that the B-subunit was sufficient for the binding and growth clearance phenotype. This was unexpected, since the B-subunit of these toxins, while crucial for adhesion and internalization by eukaryotic cells, as well as for immune activation in humans and animals, is not toxic to eukaryotic tissues without the enzymatically active A-subunit[45]. In contrast, the antibacterial effect we observed appears to be mediated through binding by the toxin. It is important to note however, that we have not excluded the possibility that the A and B subunits could have a synergistic effect to cause increased clearance when the two are together in the holotoxin configuration.

The fact that the binding subunits of CTB and LTB alone are sufficient for clearance shows that the ADP-ribosylating action of the toxins is not necessary for the activity. The observations that this exposure does not alter individual cell morphology (Supplementary Fig. 2A-B) and that CTB does not clear *C. jejuni* if added after cells have grown (Supplementary Fig. 2) suggest

that the observed effect is bacteriostatic as opposed to bactericidal. This is in contrast to lysis buffer, which causes visible clearing within minutes of spotting onto *C. jejuni* (Supplementary Fig. 2D). In *Cryptococcus neoformans*, growth inhibition has been observed as a result of antibodies directed against its capsular polysaccharide; the antibodies completely surround the cells and prevent it from dividing by encapsulating the organism[46]. However, if our observed CTB-induced clearing occurred via the same mechanism, we would expect α-GM1 antibodies to reproduce the clearing effect, which we did not observe (Supplementary Fig. 2). Since the outer leaflet of the outer membrane acts as a permeability barrier, helping bacteria to regulate flow of harmful compounds, as well as beneficial nutrients[47], we postulated that disruption of this barrier may cause the bacteriostatic effect that we observe. Indeed, the results of our EtBr accumulation assay suggest that the mechanism is related to an increase in membrane permeability upon toxin binding. To determine if increased EtBr accumulation was an effect of the toxin blocking active efflux of EtBr, a mutant in the periplasmic component of the *Campylobacter* multidrug efflux pump, *cmeA* was inactivated. This mutant still showed an increase in EtBr permeability even in the absence of an active efflux pump, suggesting that CTB might act directly on the membrane. Interestingly, although CTB bound extensively to *E. coli* CWG 308 pGM1/pCst displaying GM1 ganglioside mimics on its LOS, none of the toxin subunits or the CT holotoxin were able to clear this engineered strain (Supplementary Fig. 1) and the EtBr accumulation assay showed no effect of CTB on membrane permeability. This could be due to inherent differences between these bacteria in their membrane architecture.

*C. jejuni* is known to stochastically vary expression of its surface structures through changes in the homopolymeric nucleotide tract length within specific genes resulting in frequent frameshifts and premature stop codons during replication. This effectively leads to "on/off-switching" of genes containing these tracts, many of which are involved in surface glycan biosynthesis and transfer affording *C. jejuni* the ability to rapidly shift the composition and diversity of surface structures in that community. Alterations in these structures has been shown to reduce the susceptibility of *C. jejuni* to serum, bacteriophages[48] and antimicrobial peptides[37,38]. As well, even small changes in LOS have been shown to have a profound effect on *C. jejuni* invasiveness[38]. By encoding a polyguanosine tract in the *cgtB* gene, strain *C. jejuni* 11168 varies the expression of the galactosyltransferase responsible for addition of the LOS terminal galactose. Those cells expressing "off-switched" *cgtB* will thus present the GM2 ganglioside epitope as opposed to that of GM1. This study demonstrates that LOS from CTB-exposed cells is substantially reduced in CTB binding, and that these cells display off-switched *cgtB* expression at the poly-G tract. Together these results indicate that CTB and LTB select against GM1 epitope expression by *C. jejuni*. Interestingly, we also found that pre-incubation with CTB for less than two cell divisions was sufficient to decrease the virulence of *C. jejuni* in a *G. mellonella* model. This not only suggests that expression of the GM1 ganglioside in wax moth larvae is important for virulence, but also shows that CTB binding to *C. jejuni* has consequences on the biology of the organism that extend beyond growth inhibition *in vitro*.

Since our studies demonstrated that there is an abundance of ganglioside structures in the chicken intestine, it was unexpected, but reasonable to find many ganglioside-mimicking bacteria existing in this environment. Investigation of chicken cecal cross-sections identified bacteria that were readily labeled by both CTB and by α-GM1 antibodies, indicating the presence of other ganglioside receptors and possibly fucosylated structures. Additionally, the CTB bound abundantly to the mucus and goblet cells, which were not bound by α-GM1 antibody. This binding

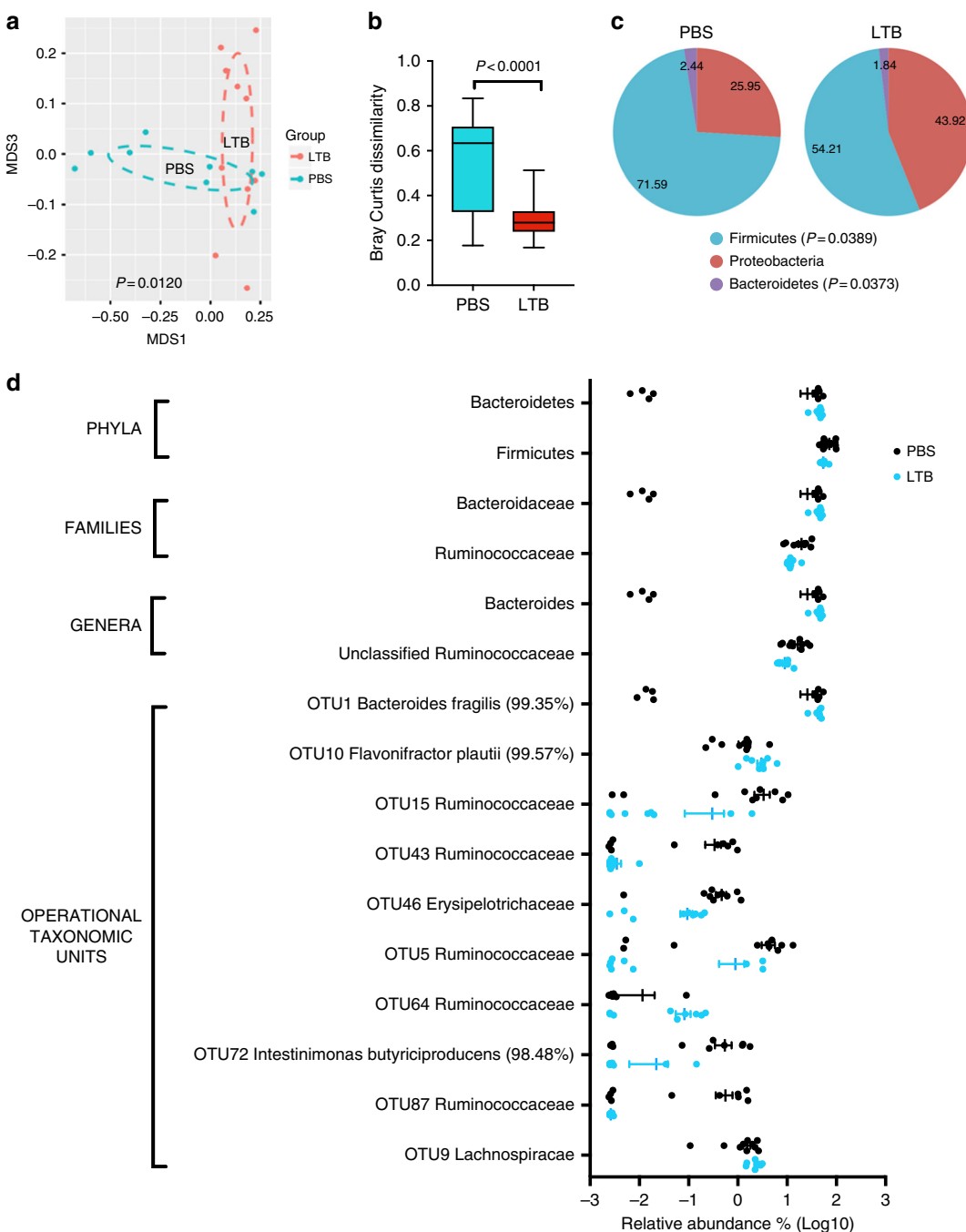

**Fig. 7** AB$_5$ toxins cause shifts in the gut microbiome of chickens by affecting diversity and composition. **a** NMDS plot based on Bray Curtis distances show that introduction of heat-labile enterotoxin (LTB, $n = 9$) causes a significant shift ($p = 0.0120$, Adonis PERMANOVA test) in the gut microbiome of chickens relative to chickens treated with PBS ($n = 10$). **b** Although α-diversity is not affected, LTB decreases β-diversity ($p < 0.0001$, two-tailed unpaired $t$-test with Welch's correction) indicating that the toxin lowers inter-individuality of the gut microbiota. **c** LTB decreases the relative abundance of Firmicutes ($p = 0.0389$) with parallel increases in Bacteroidetes ($p = 0.0373$). **d** Significant changes in relative abundance (expressed as log$_{10}$) at selected taxonomic level (see Table 2 for $p$-values and percentages of relative abundance). Whiskers in **b** represent minimum and maximum values. Lines and bars in **d** represent means and standard error of means. Source data are provided as a Source Data file

was largely abrogated by the addition of an α1-2-fucosidase to cross-sections prior to fluorescence detection, indicating that CTB binding to these structures was mediated through its fucose-binding domain. In support of this hypothesis, fucose is known to be abundant in mucins and in goblet cells producing mucin[49]. Remarkably, both CTB and the α-GM1 antibodies also bound extensively to bacteria in the intestinal lumen, suggesting that there are other bacteria present in the chicken gut that display antigens resembling GM1 gangliosides. Ganglioside mimicry is a

phenomenon thought to be rare in bacteria, and as described earlier, can lead to autoimmune disease in humans. Therefore, the observation that other ganglioside-mimicking bacteria are prevalent in food animals warrants further investigation into their identity and into the nature of the glycoconjugates they display.

Given the broad distribution of GM1 ganglioside-mimicking bacteria among the gut microbiota of chickens, we hypothesized that the administration of AB$_5$ toxins would alter the microbial

**Table 2 Relative abundance and *p*-values of statistically significant changes in taxa caused by administration of LTB**

| Taxonomic level | Name | PBS (%) | LTB (%) | *p*-value* |
|---|---|---|---|---|
| Phylum | Bacteroidetes | 25.9458 | 43.9206 | 0.0373 |
| Phylum | Firmicutes | 71.5893 | 54.2123 | 0.0389 |
| Family | Bacteroidaceae | 25.9458 | 43.9206 | 0.0373 |
| Family | Lachnospiraceae | 42.8754 | 31.8613 | 0.0507 |
| Family | Ruminococcaceae | 19.529 | 12.2352 | 0.0197 |
| Genus | *Bacteroides* | 25.9458 | 43.9206 | 0.0373 |
| Genus | unclassified_Ruminococcaceae | 16.4285 | 8.9802 | 0.0096 |
| OTU1 | *Bacteroides fragilis* (99.35%) | 25.7470 | 42.5499 | 0.0478 |
| OTU10 | *Flavonifractor plautii* (99.57%) | 1.4066 | 3.0087 | 0.0214 |
| OTU15 | Ruminococcaceae | 3.3204 | 0.3017 | 0.0247 |
| OTU43 | Ruminococcaceae | 0.3348 | 0.0034 | 0.0168 |
| OTU46 | Erysipelotrichaceae | 0.4693 | 0.0941 | 0.0064 |
| OTU5 | Ruminococcaceae | 4.3303 | 0.8824 | 0.0302 |
| OTU64 | Ruminococcaceae | 0.0116 | 0.0822 | 0.0198 |
| OTU72 | Intestinimonas Butyriciproducens_(98.48%) | 0.5461 | 0.0219 | 0.0266 |
| OTU87 | Ruminococcaceae | 0.5623 | 0.0026 | 0.0214 |
| OTU9 | Lachnospiraceae | 1.5424 | 2.3670 | 0.0227 |

*Statistical significance obtained using two-tailed unpaired *t*-test with Welch's correction

composition. The most striking impact we observed was the decrease in microbes in the Firmicutes phylum, which represents a major component of the chicken gut microbiome. We were also able to isolate a member of the Firmicutes, *E. gallinarum/casseliflavus*, from the chicken gut based on its ability to mimic GM1-ganglioside. Interestingly, *E. gallinarum* has recently been associated with another autoimmune disease, systemic lupus erythematosus[50]. Other researchers have also observed gut microbiome changes following *V. cholerae* infection[51,52]. Hsiao et al. showed that *V. cholerae* infection correlated to an unhealthy microbiome, leading to long-lasting effects on gut microbial composition even after the infection had subsided[51,52] while Monira et al. linked *V. cholerae* infection to increases in Proteobacteria and decreases in other phyla, including Firmicutes[52]. In the published studies, it is not possible to distinguish between direct toxin effects on the gut microbiota vs consequences of the pathological process; however, the sole administration of the toxin B-subunits used in our studies does not induce pathology[45], suggesting that these AB$_5$ toxins could directly provide a competitive advantage for those that produce them through growth inhibition or simply reduced colonization. It may be particularly important for AB$_5$ toxin-producers to target ganglioside-mimicking bacteria due to the fitness that mimicry imparts in dealing with host defenses[37,38]. This would make AB$_5$ toxins particularly useful tools in the microbiome and supports the notion that these toxins did not only (or primarily) evolve to target the host, but instead increase competitiveness of the producers as colonizers of the gastrointestinal tract.

The evidence that other gut bacteria have the capacity to mimic GM1 gangliosides and that AB$_5$ toxins may influence these species has other implications for human health. As mentioned above, ganglioside mimicry is closely linked to the development of GBS, but not all persons infected with GM1-mimicking *C. jejuni* develop the disease. It is possible that other ganglioside-mimicking bacteria could be present in our gut or appear transiently through ingestion of poultry products and could impact individual disease susceptibility by either immune tolerizing or training. Further studies into the extent to which other bacteria presenting these antigens contribute to the manifestation of GBS are warranted. It is also relevant that CTB is currently administered as part of an increasing number of health-promoting strategies including use in the oral cholera vaccine, as a toxoid

adjuvant (with unexpected cross-reactivity to the *C. jejuni* major outer membrane protein[53]), and recently as an anti-inflammatory agent for treatment of diseases such as diabetes mellitus and atherosclerosis[54]. Taken together, our studies identify a new role for AB$_5$ toxins and may explain why some organisms have developed mechanisms to vary their ganglioside mimics. We demonstrate that the chicken gut is rich in ganglioside structures derived from the host and the resident microbiota may impact both bacterial competition and human health.

## Methods

**Bacterial strains and growth conditions.** The *C. jejuni* strains used in this study were grown on NZCYM agar (Difco) under microaerobic conditions (85% N$_2$, 10% CO$_2$ and 5% O$_2$) at 37 °C. The *E. coli* strains were grown on LB agar under atmospheric conditions at 37 °C. Mutant strains were grown in media supplemented with ampicillin (50 μg mL$^{-1}$) and/or kanamycin (25 μg mL$^{-1}$). Table 1 lists the bacterial strains used in this study along with their sources.

To create the *C. jejuni* 11168 Δ*cgtB* mutant, primers cgtB-mut-F-XhoI and cgtB-mut-R-SacI (Supplementary table 1) were used to PCR-amplify the *cgtB* gene from *C. jejuni* 11168. The fragment was then ligated into pGEM®-T Easy Vector (Promega) and primers InvPCR-cgtB-F-BamHI and InvPCR-cgtB-R-BamHI (Supplementary table 1) were used to perform inverse PCR to introduce a *BamHI* site into *cgtB*. The kanamycin cassette was isolated from the pMW2 plasmid by *BamHI* digest and inserted into *cgtB*. In the case of the Δ*neuC1* mutant, primers neuC1F15 and neuC1R984 (Supplementary table 1) were used to amplify the gene out of *C. jejuni* 11168. The PCR product was ligated into pPCR-Script Amp and a kanamycin resistance cassette from pILL600 was inserted into the *NdeI* restriction site present within the gene. Next, for both mutants, *C. jejuni* 11168 cells were grown on kanamycin (50 μg mL$^{-1}$) selective agar. The resulting colonies were evaluated for insertion of the kanamycin cassette by PCR and sequencing. For the Δ*cmeA* mutant, plasmid pET22b Cf-CmeA[55] was linearized using *SpeI* and blunt-ended with T4 DNA polymerase. The *cat* cassette was obtained from plasmid pRY109 after *SmaI* digestion. After ligation, the correct orientation of the cassette in *cmeA* was confirmed by sequencing of candidate plasmids with primers CmeA-F and CmeA-R (Supplementary table 1). These primers were then used to amplify *cmeA::cat* by PCR. The purified PCR product was used to transform strain 11168 by natural transformation. CmR clones were analyzed by PCR for the correct insertion of the cassette into the chromosome with CmeA-os-R and CmeA-os-F (Supplementary table 1). Absence of CmeA in the mutant was also confirmed by western blotting with CmeA-specific antibodies.

**Immunogold labeling and transmission electron microscopy.** Following growth of *C. jejuni* or *E. coli* overnight, cells were harvested in NZCYM or LB broth, respectively, and their optical density at 600 nm (OD$_{600}$) was adjusted to 0.5 for *C. jejuni* or 1.0 for *E. coli*. Then, 2 μL of CTB (1 mg mL$^{-1}$, Sigma) was added to 2 mL of each cell suspension. The suspensions were incubated under normal growth conditions for each organism for 1 h and then centrifuged at 6,000 × *g* for

4 min Cell pellets were washed and resuspended in PBS. Immunogold labeling and transmission electron microscopy were carried out as described in previous reports with minor modifications[19,56]. Briefly, Formvar-coated copper grid was floated on top of the cell suspension for 1 h. Grids were then floated on blocking solution (PBS with 5% bovine serum albumin) for 1 h to block, followed by 1:50 rabbit α-CT antibodies (Fitzgerald Industries International) (in blocking solution) for 1 h, and goat α-rabbit IgG conjugated to 10-nm gold particles (BB International, diluted 1:50 in blocking solution) for 2 h, washing 3 times in blocking solution after each step. The grids were then examined by transmission electron microscopy (Philips Morgagni 268; FEI Company) and images were taken using a charge-coupled camera and controller (Gatan) and processed using DigitalMicrograph (Gatan).

**Bacterial clearance assay.** The effect of the holotoxins and toxin subunits on bacterial growth was examined by spotting CT, CTA, CTB or LTB onto cell-containing double layer agar plates prepared using the standard overlay agar method commonly used in bacteriophage plaque assays[57]. *C. jejuni* HS:19, 11168, and HS:3 strains or *E. coli* CWG 308 and CWG308 pGM1a/pCst were grown overnight and harvested in NZCYM or LB broth and adjusted to an OD600 of 0.1. Then, 200 μL of this suspension was mixed with 5 mL of 0.6% molten NZCYM or LB agar and poured onto a previously solidified and pre-warmed NZCYM or LB agar plate containing 1.5% agar. The molten agar layer was allowed to solidify at RT for 15 min and then 2–5 μL of toxin (1 mg mL$^{-1}$), or toxin mixed 1:1 with competitive binding glycan (both at 1 mg mL$^{-1}$) in the case of competitive inhibition assays, was spotted onto the agar surface. For the concentration determination assay, CTB was diluted with Milli-Q H$_2$O to contain 5, 1, 0.5, or 0.25 μg in 5 μL spotted onto *C. jejuni* HS:19. These plates were incubated agar-side-down for 18–24 h under normal growth conditions as described above and imaged using a GO21 camera connected to an Olympus SZX16 stereoscope.

**Scanning electron microscopy.** Following a clearance assay with CTB on *C. jejuni* HS:19 as described above, agar squares were excised from inside, outside or at the interface of the clearance zones and prepared using the following established protocol[58]. The squares were excised using a scalpel and trimmed to leave only the top layer of agar (about 2 mm) and then incubated in scanning electron microscopy (SEM) fixative (2.5% glutaraldehyde; 2% paraformaldehyde in 0.1 M phosphate buffer) overnight at 4 °C. Excised squares were then washed in 0.1 M phosphate buffer three times for 10 min and dehydrated using a series of 15 min washes: 50% ethanol, 70% ethanol, 90% ethanol, 100% ethanol, ethanol: hexam-ethyldisilazane (HMDS) 75:25, ethanol:HMDS 50:50, ethanol:HMDS 25:75 and 100% HMDS. After leaving HMDS to evaporate overnight, the excised squares were mounted onto SEM stubs and sputter-coated with the Hummer sputtering system (Anatech Ltd.). The squares were then imaged using the Phillips/FEI (XL30) scanning electron microscope (Philips/FEI) with an electron beam energy of 20 kV.

**Isolation of *C. jejuni* LOS from clearance zones.** Following a clearance assay with CTB or LTB, sections of agar were excised using a scalpel from either inside or outside the observed clearance zone, and the agar sections were incubated for 15 min at RT to elute the embedded cells. The eluate was then transferred to a new tube and centrifuged at 6000 × *g* for 4 min to collect the cells. The pellet was resuspended in 100 μL of PBS, spread onto NZCYM plates and incubated under microaerobic conditions at 37 °C overnight. The cells were then harvested in 500 μL PBS, centrifuged at 18,300 × *g* for 90 s and resuspended in 50 μL sample buffer (100 mM Tris-Cl (pH 8.0), 2% β-mercaptoethanol, 4% SDS, 0.2% bromo-phenol blue, 0.2% xylene cyanol, 20% glycerol). This mixture was boiled for 10 min at 95 °C and cooled to RT before adding 47.5 μL additional sample buffer with 2.5 μL proteinase K (20 mg mL$^{-1}$) and incubating overnight at 37 °C. The following day, the proteinase K was inactivated by heating to 65 °C for 1 h and the samples were either loaded directly onto an SDS-PAGE gel or stored at −20 °C.

**Silver staining of LOS.** After isolation of LOS, the samples were separated by SDS-PAGE using 15% SDS-PAGE. The gel was then silver-stained according to the method described by Tsai and Frasch (1982) with a few modifications[59]. The separated gels were immediately immersed in fixing solution (55% Milli-Q H$_2$O, 40% ethanol, 5% acetic acid) and incubated for at least 1.5 h. The gels were then rinsed 4 times and washed twice for 5 min in Milli-Q H$_2$O before being placed in oxidizing solution (fixing solution with 0.7% periodic acid) for 10 min After rinsing 4 times and washing twice for 5 min in Milli-Q H$_2$O, the gels were immersed in pre-stain solution (18.67 mM NaOH, 1.3% NH$_4$OH in Milli-Q H$_2$O) for 10 min, then stained with staining solution (pre-stain solution with 39.2 mM silver nitrate). Following 4 rinses and 2 washes for 10 min in Milli-Q H$_2$O, development was done using the BioRad Silver Stain Developer as directed and development was stopped by adding stopping solution (5% acetic acid in Milli-Q H$_2$O).

**Far western blot of LOS and *cgtB* sequencing.** After isolation of the LOS as described above, the samples were separated by SDS-PAGE and the gel was transferred to a PVDF membrane overnight at 30 V at 4 °C. The membrane was then blocked with 5% skim milk/PBST for 1 h, probed with CTB (1:100,000) for 1 h, washed 3 × 10 min in PBST, probed with rabbit α-CT antibodies (1:6,500) for

1 h, washed 3 × 10 min PBST, and probed with goat α-rabbit-HRP antibodies (1:20,000) for 1 h. The membrane was developed using a chemiluminescent substrate mix (Western Lightning™ Plus-ECL from PerkinElmer) and images were captured using X-ray development film (Kodak).

Cells were isolated from inside and outside the clearance zones of agar using the same method as was described for the isolation of LOS. For DNA isolation, once the isolated *C. jejuni* 11168 cells were grown on NZCYM, cells were resuspended in 500 μL of TE Buffer (10 mM Tris- HCL pH 8.0, 0.1 mM EDTA in Milli-Q H$_2$O) and centrifuged at 18,300 × *g* for 2 min. The pellet was resuspended in 0.2 mL Buffer TE and genomic DNA was isolated using the Genomic DNA Purification Kit (Fermentas) according to manufacturer's instructions. The *cgtB* gene was amplified using Vent polymerase (NEB) and the primers described by Linton et al.:[22] DL39 and DL41 (Supplementary table 1). The resulting solution was purified using the GeneJET™ Gel Purification Kit (Fermentas) as instructed by the manufacturer. The purified PCR product was then sequenced using the reverse primer 1139R (Supplementary table 1) with sequencing services offered by the Molecular Biology Service Unit at the University of Alberta.

**Ethidium bromide accumulation assay.** The ethidium bromide (EtBr) accumulation assay was performed using the following established protocol[30]. *C. jejuni* 11168 wildtype, Δ*cgtB*, Δ*cmeA*, Δ*neuC1* cells, as well as *E. coli* CWG 308 and pGM1/pCst were harvested from agar plates in PBS and adjusted to OD600 = 0.2. The cells were then incubated with CTB (50 μg mL$^{-1}$) for 30 min at 37 °C before adding EtBr to a final concentration of 1.875 μg mL$^{-1}$. The accumulation of EtBr was measured by relative fluorescence every 2 min over a total of 20 min using the Bio Tek Synergy H1 plate reader with an excitation wavelength of 530 nm and emission wavelength of 600 nm. This experiment included 7 biological replicates for 11168 wildtype, 4 for Δ*cgtB*, Δ*cmeA* and Δ*neuC1*, and 3 for both of the *E. coli* strains. Each biological replicate included 3 technical replicates.

**Galleria mellonella infection assay.** The *Galleria mellonella* wax moth model was used as a model to assess *C. jejuni* virulence similar to what has previously been done[60]. *C. jejuni* HS:19 was suspended in 10 mM MgSO$_4$ solution to an OD600 = 1.0 (~1.0 × 10$^9$). The cell suspensions were then mixed with either MgSO$_4$, or CTB to a final concentration of 50 μg mL$^{-1}$, and incubated for 4 h at 37 °C. Following this incubation, cell counts were verified by serial dilution plating and 5 μL of suspension or control MgSO$_4$ was injected into the hind leg of the wax moth larvae (Backwater Reptiles Inc.). The larvae were incubated at 37 °C for 5 days before counting the proportion of dead and alive moths. The *G. mellonella* incubation time coincides with growth curves established to determine time needed for *C. jejuni* HS:19 to cause mortality. This experiment included 3 biological replicates for both HS:19 and HS:19 + CTB, 2 for MgSO$_4$ and 1 for CTB. Each biological replicate included 3 technical replicates.

**Chicken studies.** Commercial broiler chickens (Ross 308, Avigen) were obtained on the day of hatch from Lilydale, Edmonton. Upon arrival, birds were divided into groups with up to 9 chickens each. The birds were orally gavaged with PBS, PBS + CTB, or PBS + LTB initially on day 30, then again 8 h and 12 h later using 100 μg of toxin per 300 μL gavage. The birds were then euthanized on day 31 and both ceca were removed and processed immediately. All animal studies were carried out in accordance with the protocol approved by the Animal Care and Use Committee at the University of Alberta, protocol number AUP00003.

**Fluorescence microscopy.** Intact chicken ceca were soaked in fresh Carnoy's fixative (60% ethanol, 30% chloroform, 10% glacial acetic acid) for 2 h and then moved to 100% ethanol. They were then cut into appropriate sized cross-sections with a scalpel, anchored in tissue cassettes and stored in 70% ethanol prior to embedding. Paraffin embedding involved sequential ethanol dehydration steps: 1 h in 70% ethanol, 1 h in 90% ethanol, 1.5 h in 100% ethanol, 1.5 h in 100% ethanol, 1.25 h in ethanol:toluene (1:1), followed by 2 × 0.5 h in 100% toluene. The specimen was then put in two wax treatments for 2 h each and embedded the following morning in paraffin.

Immunofluorescent staining of the paraffin-embedded sections was conducted using the following established protocols[61.] Tissue sections were deparaffinized using 5 min washes in xylene (4×), 100% ethanol (2×), 95% ethanol, 70% ethanol and dH$_2$O. Antigen retrieval was performed by steaming slides for 30 min in sodium citrate buffer (10 mM Sodium Citrate, 0.05% Tween 20, pH 6.0). Slides were blocked for 1 h using blocking buffer (PBS containing: 0.2% Triton X-100, 0.05% Tween, 1% bovine serum albumin, 5% donkey serum). Where indicated, slides were treated with α1-2-fucosidase (NEB, P0724S) following antigen retrieval in sodium citrate buffer, but prior to blocking with donkey serum buffer. Slides were treated for 1 h, with 1 μl (20 units) of α1-2-fucosidase in 10 μl of 1× Glycobuffer (supplied by the manufacturer), at 37 °C, followed by a PBS wash. Each slide was then treated with CTB (1 mM, Sigma, Cat. No. C9903) for 1 h, followed by either α-CTB antibody (1:100) or α-GM1 antibody (1:100, Abcam) for 1 h. Slides were visualized using Alexa Fluor 488-conjugated donkey α-rabbit IgG (1:1000, Invitrogen). The tissues were mounted using ProLong Gold antifade reagent containing DAPI (Invitrogen). The stained slides were viewed using a Zeiss

AxioImager Z1, and photographed using an AxioCam HRm camera with AxioVision software.

Whole cell surface staining of GM1 pull-down candidates was performed as follows: isolated bacteria were harvested from their respective growth media and adjusted to $OD_{600} = 1.0$. Cells from 100 μl were pelleted by centrifugation (1 min, 13,000 rpm), suspended in 1 ml blocking solution (PBS, 5% skim milk) and incubated on ice for 30 min Cells were subsequently probed with either α-GM1 (1:1000 in PBS; Abcam, Cat. No. 23943) or CTB (1 mM) in combination with α-CTB (1:1000 in PBS; Biorad, Cat. No. 2060–0020) and Alexa Flour-488 conjugated α-rabbit antiserum (1:500 in PBS; Invitrogen, Cat # A-11008). Each incubation was done on ice for 20–30 min and three washes with 1 ml PBS were performed in between antibody incubations. Cells were counterstained with DAPI (1 μg mL$^{-1}$), mounted on glass slides and analyzed on a DeltaVision fluorescent microscope equipped with a sCMOS camera.

**16S rRNA sequence analysis of chicken ceca**. DNA isolation and 16S rRNA tag sequence analysis of bacterial DNA isolated from chicken cecal samples was performed using the following established protocol[62]. DNA was amplified using the 926F and 1392R universal primers (Supplementary table 1) that target the V6 to V8 region of the 16S ribosomal RNA gene. The PCR conditions are described in 16S Metagenomic Sequencing Library Preparation (Illumina®, San Diego, CA). Sequencing of the 16S rRNA gene was completed with an Illumina MiSeq Sequencer using MiSeq Reagent kit V3 (Illumina). Reads were trimmed to 290 bp with the FASTX-Toolkit (http://hannonlab.cshl.edu/fastx_toolkit/), and paired-end reads were merged with the merge-illumina-pairs application (https://github.com/meren/illumina-utils/). Sequences between 440 and 470 nucleotides long were selected for analysis. To standardize the sequencing depth across samples, sequences were subsampled to 45,000 reads using Mothur v.1.35.1. Subsequently, USEARCH v8.1.1861 was used to generate operational taxonomic units (OTUs) with a 98% similarity cut-off. OTU generation included the removal of putative chimeras identified against the Gold reference database, in addition to the chimera removal inherent to the OTU clustering step in UPARSE. The resulting reads were assigned to different taxonomic levels from Phylum to Genus using a parallelized version of CLASSIFIER (rdp_classifier_v2.10.1) from the Ribosomal Database Project (RDP). Taxonomic assignment for the OTUs was obtained using the "Identify" function in EZbiocloud[63]. Sequences whose identity was lower than 97% were classified using the Classifier and SeqMatch functions in Ribosomal Database Project webpage[64,65]. The number of reads in each taxonomic bin was normalized to the proportion of the total number of reads per each sample for statistical analyses. To achieve normality for the microbiome data that were not normally distributed, values were subjected to $log_{10}$ transformations. Diversity analyses were done using MacQIIME version 1.9.1[66].

**Identification of GM1-mimicking bacteria from ceca**. Chicken cecal contents were resuspended in PBS to a concentration of 1 mg mL$^{-1}$. To address non-specific binding, the mixture was further diluted 1:5 with PBS-T containing 2% skim milk and incubated in a gravity flow column with 500 μL protein G agarose (Sigma) for 10–15 min at room temperature. Samples were mixed every 2–3 min, the flow-through was collected and incubated with 5 μL of α-GM1 antibody on ice for 15 min, mixing regularly. The suspension was then passed through a gravity flow column containing 500 μL of protein G agarose. After two washes with PBS-T, 2% skim milk and PBS-T and one final wash with PBS (5 mL each), protein G agarose beads were recovered with 1 mL PBS and 50 μL of a 5-fold serial dilution series were plated on either Brain Heart infusion (BHI), De Man, Rogosa and Sharpe (MRS), or Yeast, Casitone-Fatty Acid (YCFA) agar. Plates were incubated micro-aerobically or anaerobically at 37 °C for up to 72 h. Colonies were picked, isolated and tested for GM1 ganglioside antibody reactivity by whole cell ELISA followed by GM1 and CTB binding by fluorescence microscopy. One GM1 and CTB-reactive bacterial species was further identified by 16S rRNA gene sequencing with oligonucleotides 926R, 515F, 1068F, 1391R, and 519R (Supplementary table 1) after PCR-amplification of the 16S rRNA gene from extracted DNA with oligonucleotides 27F and 1492R (Supplementary table 1).

**GM1 antibody and lysis solution spot assay**. These spot assays were performed in the same way as the others with minor changes. For the GM1 ganglioside antibody spot assay, 2 μL of the antibody (Abcam) or 2 μg of CTB was spotted directly onto the cells prior to growth. For the lysis buffer spot assay, 2 μL of lysis buffer (undiluted from Thermo Scientific GeneJET Plasmid Miniprep Kit) or 2 μg of CTB was spotted onto cells after growth for 24 h.

**Statistics**. For the EtBr accumulation and *Galleria mellonella* data, two-tailed, unpaired *t*-tests were performed to compare the relative fluorescence upon addition of EtBr, and *C. jejuni* HS:19 infection with and without CTB respectively. For microbiome analyses, two-tailed Mann Whitney tests were performed for the CTB dataset comparisons, while two-tailed unpaired *t*-tests with Welch's correction or multiple *t*-tests were performed for the LTB comparisons using Graph Pad Prism version 7. The Adonis PERMANOVA test was performed on NMDS plots using the statistical software R version 3.3.1 (http://www.r-project.org/).

**Data availability**

All relevant data are available from the corresponding author. The source data underlying Figures 4a-b, 5a-b and 7, and Supplementary Figures 1c-d, 3 and 4 are provided as a Source Data file. The 16S rRNA datasets generated and analyzed during these studies are available in the NCBI SRA repository under BioProject PRJNA522915; the BioSamples accession numbers are listed in Supplementary table 2.

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

## Acknowledgements

We would like to thank Brandi Davis and Yee Ying Lock for preparation of samples for 16S rRNA sequencing, Arlene Oatway for help with preparing cecal samples for electron microscopy, the Molecular Biology Services Unit at the University of Alberta for performing the Sanger sequencing reactions and Drs. James and Adrienne Paton for providing us with their *E. coli* CWG 308 wildtype and pGM1/pCst strains. We also thank Dr. Xiaoming Bian and Dr. Hanwen Huang for statistical advice, Dr. Xiaorong Lin for the use of her stereoscope, and Dr. Stefan Pukatzki and Dr. Michel Gilbert for helpful discussions. This study was funded in part by fellowships to RTP from the Natural Sciences and Engineering Research Council of Canada and Alberta Innovates Health Solutions. C.M.S. is an Alberta Innovates Strategic Chair in Bacterial Glycomics. J.W. is a CAIP Chair in Nutrition, Microbes, and Gastrointestinal Health. B.A.V. is the CH.I.L.D. Foundation Chair in Pediatric Gastroenterology.

## Author contributions

R.T.P., M.S., M.E.P., H.N., C.Q.W. performed the experiments. All authors contributed to the writing of the paper. R.T.P., J.S., and C.M.S. conceived the original idea. C.C., J.W., B.A.V., and C.M.S. supervised the project.

## Additional information

**Competing interests:** The authors declare no competing interests.

