## [Peer Review File · Nature Communications]

Reviewers' comments:

Reviewer #1 (Remarks to the Author):

This exciting manuscript from Patry et al. proposes and provides strong experimental support for a fundamentally new idea about the role of glycan recognition by bacterial AB5 toxins. AB5 toxins are produced by a variety of pathogenic bacteria, and have been characterized in terms of their ability to intoxicate mammalian cells and cause disease. This manuscript demonstrates that at least two of these AB5 toxins also have the ability to bind to glycan structures on bacterial species that compete for the same niche as the AB5-producing organism. Further, AB5 toxins are bacteriostatic to some competing species, and can alter the microbiome of the host. The data presented in Figure 4 are particularly compelling, and show that growing the *C. jejuni* phase-variant strain 11168 in the presence of cholera toxin subunit B (CTB) leads to a previously-characterized genetic change that abolishes production of the GM1-like LOS glycan. The key claims are further buttressed by measuring the effect of CTB on *C. jejuni* virulence in the *G. mellonella* model and the effect of LTB on the gut microbiome of chickens. The results presented in this manuscript are important for considering human disease pathogenesis and recovery, and suggest the provocative idea that bacterial toxins may have evolved to recognize other bacteria, not (solely) the host. The experiments are carefully performed and appropriate controls are included. However, the wording of some ancillary claims may need to be adjusted to match the data presented.

1. (most important) The authors make strong claims that the effects of the toxins on *C. jejuni* and other competing species are due solely to the B subunits of the toxins, with the catalytic A subunits having no additional activity in the proposed inter-bacterial warfare. Most of the experiments presented in the manuscript were performed with just the B subunit, supporting the idea that the B subunit carries the bacteriostatic activity and also can induce selective pressure on the microbiome. However, the authors present very little data for experiments that use the holotoxin, so it is difficult to eliminate the possibility that the A subunit might have some additional effect. Figure 2D is the main piece of data that compares the holotoxin to CTB. In line 352, the authors make the claim that the clearance caused by CTB was comparable to that caused by the holotoxin, but the data presented are not quantitative. Further, the clearance due to CT actually looks a little stronger than the clearance due to CTB. My recommendation is that the authors modulate their claims to indicate that the B subunits (of CT and LT) are sufficient to mediate the effects examined here, but that the current data do not exclude the possibility that the A subunit could have some additional effect.
2. In describing the data presented in Figure 4, panels A and B, the authors make the proposal that the doublet of bands shown in the first lane of the silver stain gel corresponds to full-length LOS and LOS that lacks the terminal Gal. The absence of the higher molecular weight band in lanes 2 and 3 is attributed to incomplete assembly of the CTB recognition epitope in the absence of Gal. However, in the Western blot (Fig 4B), the species recognized by CTB forms a much more diffuse "smear." Can the authors provide a more full explanation of why the band on the Western blot might be so much more broad than the band on the silver-stained gel? Also, can the authors provide molecular weight information or mobility standards for these gels?
3. In line 542, the authors use the observation that the *cmeA* mutant has no effect EtBr permeability to make the claim that CTB is acting directly on the membrane. This is speculative, given the limited data presented. Suggest revising to "This mutant still showed an increase in EtBr permeability even in the absence of the known efflux pump, suggesting that CTB might act directly on the membrane."
4. For the *Enterococcus gallinarum/casseliflavus* bacteria shown in Figure 6, panels G and H, can the authors provide any data on what fraction of the cells are CTB and GM1 positive? It is difficult to tell from the images.

Smaller points:

Please provide information on the anti-GM1 antibody (vendor and order number) used in these studies.

The authors may wish to double-check the wording of the two sentences from lines 509 – 514. It seems like some words may have been left out in the final round of edits.

In the legend to Figure 5, the wording of the final sentence (information about the number of replicates) is unclear.

Figure 6: suggest adding labels to the images indicating which antibody (anti-GM1), detection reagent (CTB), and/or other treatment (fucosidase) has been applied. This would make it easier for the reader.

Reviewer #2 (Remarks to the Author):

This manuscript describes a new function of the AB5 toxins (CT and LT) in inhibiting the growth of *Campylobacter jejuni* and other GM1 ganglioside mimicking micro-organisms in the chicken gut. The authors demonstrated that the inhibitory effect was due to the B subunit of the AB5 toxins, which binds to the GM1-mimicking lipooligosaccharides on the surface of *C. jejuni* and increases permeability of the cell membrane. Furthermore, the authors found that exposure of chickens to CTB resulted in shifts in the gut microbial composition and suggested that these toxins may have new functions in modulating bacterial competition in the gut.

Overall the authors did a nice job demonstrating that the interaction of AB5 toxins with *Campylobacter*, producing a bacteriostatic effect on the growth of the organism. It is an interesting story that suggests that the AB5 toxins may contribute to bacterial competition in the gut beyond causing diarrhea in human. This reviewer has a few comments for the authors to consider.

1. The bacteriostatic effect of the B subunit could be measured in a quantitative way by determining the minimal inhibition concentrations in HS:19 and HS:3 as well as the *cgtB* mutant strain. This data would complement the qualitative results presented in Figure 2 and further strengthen the conclusion.

2. The EB accumulation could be improved by measuring multiple time points and by including a complemented strain of the *cgtB* mutant. Given that it is a key experiment determining how the toxin affects *Campylobacter* growth, it should be designed more rigorously. Also, both the method and the legend of figure 5 did not describe how the percent increase in fluorescence was calculated. This should be clearly explained in the method section. Additionally, panel B should be indicated in line 680 (before *G. mellonella*).

3. LTB was found to cause population shifts in the intestinal microbiota of chickens. The authors suggested that it was because the toxin inhibited bacterial growth. Since the authors have identified more GM1-mimicking bacteria from chicken guts such as *Enterococcus gallinarum*, it will be interesting to see if the toxin also have any inhibitory effects on these GM1-mimicking bacteria in culture media. This would demonstrate the broader impact of the toxins and further strengthen the conclusion of the study.

4. The image quality for Fig. 2 (D, E, and F; clearance assay) could be improved. The ones for 11168 were especially unclear.

5. Line 33: "CTB" and "LTB" should be defined. Line 133: it was the first use of "LTB" in the main text and should be defined

Author responses for manuscript NCOMMS-18-25159-T

Point-by-point response to the referees' comments and changes introduced after checklist review

Reviewers' comments:

Reviewer #1 (Remarks to the Author):

This exciting manuscript from Patry et al. proposes and provides strong experimental support for a fundamentally new idea about the role of glycan recognition by bacterial AB5 toxins. AB5 toxins are produced by a variety of pathogenic bacteria, and have been characterized in terms of their ability to intoxicate mammalian cells and cause disease. This manuscript demonstrates that at least two of these AB5 toxins also have the ability to bind to glycan structures on bacterial species that compete for the same niche as the AB5-producing organism. Further, AB5 toxins are bacteriostatic to some competing species, and can alter the microbiome of the host. The data presented in Figure 4 are particularly compelling, and show that growing the *C. jejuni* phase-variant strain 11168 in the presence of cholera toxin subunit B (CTB) leads to a previously-characterized genetic change that abolishes production of the GM1-like LOS glycan. The key claims are further buttressed by measuring the effect of CTB on *C. jejuni* virulence in the *G. mellonella* model and the effect of LTB on the gut microbiome of chickens. The results presented in this manuscript are important for considering human disease pathogenesis and recovery, and suggest the provocative idea that bacterial toxins may have evolved to recognize other bacteria, not (solely) the host. The experiments are carefully performed and appropriate controls are included. However, the wording of some ancillary claims may need to be adjusted to match the data presented.

We thank the reviewer for their enthusiasm. We have detailed the wording adjustments below.

1. (most important) The authors make strong claims that the effects of the toxins on *C. jejuni* and other competing species are due solely to the B subunits of the toxins, with the catalytic A subunits having no additional activity in the proposed inter-bacterial warfare. Most of the experiments presented in the manuscript were performed with just the B subunit, supporting the idea that the B subunit carries the bacteriostatic activity and also can induce selective pressure on the microbiome. However, the authors present very little data for experiments that use the holotoxin, so it is difficult to eliminate the possibility that the A subunit might have some additional effect. Figure 2D is the main piece of data that compares the holotoxin to

CTB. In line 352, the authors make the claim that the clearance caused by CTB was comparable to that caused by the holotoxin, but the data presented are not quantitative. Further, the clearance due to CT actually looks a little stronger than the clearance due to CTB. My recommendation is that the authors modulate their claims to indicate that the B subunits (of CT and LT) are sufficient to mediate the effects examined here, but that the current data do not exclude the possibility that the A subunit could have some additional effect.

The authors agree with the reviewer and we have changed wording in the key statements (specifically lines 127-130, 297-298, 302-304, and 306) to reflect that although the B subunits are sufficient for clearance and the A subunit is not necessary, this does not exclude the possibility of a synergistic effect of the two, causing increased clearance by the holotoxin.

2. In describing the data presented in Figure 4, panels A and B, the authors make the proposal that the doublet of bands shown in the first lane of the silver stain gel corresponds to full-length LOS and LOS that lacks the terminal Gal. The absence of the higher molecular weight band in lanes 2 and 3 is attributed to incomplete assembly of the CTB recognition epitope in the absence of Gal. However, in the Western blot (Fig 4B), the species recognized by CTB forms a much more diffuse “smear.” Can the authors provide a more full explanation of why the band on the Western blot might be so much more broad than the band on the silver-stained gel? Also, can the authors provide molecular weight information or mobility standards for these gels?

When separating glycolipids on polyacrylamide gels, it is common not to see sharp bands due the presence of the lipid in the sample. In the case of our gel, the majority of the LOS can be seen in the lower molecular weight range in the silver stained gel – however, the higher molecular weight smear is present in low amounts and is below the detection limit of the silver stain method. In contrast, the Western blot method is more sensitive due the 3-step signal amplification after subsequent binding of CTB, anti-CT and anti-rabbit antibodies. The resulting increase in sensitivity therefore detects the low amounts of the higher molecular weight smear in the Far Western blot that cannot be seen on the silver stained gel.

The molecular weight marker information was omitted since in contrast to soluble proteins, lipid-carbohydrate molecules do not migrate according to their molecular weight when separated by PAGE. However, the authors agree with the reviewer that having a molecular weight marker reference can be useful in gaining some relative perspective of the migration pattern. We have now included a line on the Western to indicate the lowest molecular weight marker signal that was visible (we do not usually include MW markers on the gels, but always include them on the Western to monitor the transfer).

3. In line 542, the authors use the observation that the *cmeA* mutant has no effect EtBr permeability to make the claim that CTB is acting directly on the membrane. This is speculative, given the limited data presented. Suggest revising to “This mutant still showed an increase in EtBr permeability even in the absence of the known efflux pump, suggesting that CTB might act directly on the membrane.”

The authors agree with the reviewer and appreciate the suggested change. The sentence has been replaced with what was suggested.

4. For the *Enterococcus gallinarum/casseliflavus* bacteria shown in Figure 6, panels G and H, can the authors provide any data on what fraction of the cells are CTB and GM1 positive? It is difficult to tell from the images.

Upon further analysis by MALDI-mass spectrometry, we determined that this isolate actually contains a mixture of both *Enterococcus gallinarum* and *Enterococcus casseliflavus* (so the *E. gallinarum/casseliflavus* description in the paper is now literally correct). This could explain why only a fraction of cells appear to be

CTB and GM1 positive. However, even to get enough DNA from the cells for sequencing required weeks of troubleshooting to break-open these Gram-positive cells without sheering the DNA and we are still doing additional sequencing runs to obtain all the genetic information. Similarly, we are optimizing our usual methods to determine which of these two isolates, or whether both, are CTB and GM1 positive. Due to some messy patterns seen by Western blot analysis on lysates from these bacteria, it is possible that the mimic is present on a glycoprotein. The only putative Enterococcal glycoprotein currently known is produced by donor cells 30-40 min following release of a pheromone by recipient cells to induce aggregation for conjugation. This could also explain why the mimics appear to be ON some of the time and OFF at other times, however all our results are too speculative to be included in the current manuscript and we feel the characterization of these isolates is beyond the scope of this paper.

Smaller points:

Please provide information on the anti-GM1 antibody (vendor and order number) used in these studies.

The vendor (Abcam) has been noted in line 610 of the methods section. We did not include the order number because multiple orders of this antibody were used to complete the described experiments.

The authors may wish to double-check the wording of the two sentences from lines 509 – 514. It seems like some words may have been left out in the final round of edits.

We thank the reviewer for noticing this, the sentence has now been corrected.

In the legend to Figure 5, the wording of the final sentence (information about the number of replicates) is unclear.

This sentence has been reworded for clarity and a more detailed description of the replicates has been added to the methods section of each experiment.

Figure 6: suggest adding labels to the images indicating which antibody (anti-GM1), detection reagent (CTB), and/or other treatment (fucosidase) has been applied. This would make it easier for the reader.

This is a great suggestion. Labels have now been added to make this figure more clear.

Reviewer #2 (Remarks to the Author):

This manuscript describes a new function of the AB5 toxins (CT and LT) in inhibiting the growth of *Campylobacter jejuni* and other GM1 ganglioside mimicking micro-organisms in the chicken gut. The authors demonstrated that the inhibitory effect was due to the B subunit of the AB5 toxins, which binds to the GM1-mimicking lipooligosaccharides on the surface of *C. jejuni* and increases permeability of the cell membrane. Furthermore, the authors found that exposure of chickens to CTB resulted in shifts in the gut microbial composition and suggested that these toxins may have new functions in modulating bacterial competition in the gut.

Overall the authors did a nice job demonstrating that the interaction of AB5 toxins with *Campylobacter*, producing a bacteriostatic effect on the growth of the organism. It is an interesting story that suggests that the AB5 toxins may contribute to bacterial competition in the gut beyond causing diarrhea in human. This reviewer has a few comments for the authors to consider.

We thank this reviewer for their nice comments.

1. The bacteriostatic effect of the B subunit could be measured in a quantitative way by determining the minimal inhibition concentrations in HS:19 and HS:3 as well as the *cgtB* mutant strain. This data would complement the qualitative results presented in Figure 2 and further strengthen the conclusion.

Since these assays can only be done on agar plates, we have modified Figure 2 to now include a new panel (H) which shows CTB spotted at different concentrations to show the range of concentrations that can be used to observe the effect.

2. The EB accumulation could be improved by measuring multiple time points and by including a complemented strain of the *cgtB* mutant. Given that it is a key experiment determining how the toxin affects *Campylobacter* growth, it should be designed more rigorously. Also, both the method and the legend of figure 5 did not describe how the percent increase in fluorescence was calculated. This should be clearly explained in the method section.

Each experiment was originally done measuring time points every 2 minutes for a total of 20 minutes. We did not observe any striking features during the experiments and thought that the data was better represented in bar form. However, we recognize the value in seeing the time course, so an additional figure has been added to the supplementary to include representative time course data for each strain.

Also, to further indicate that the binding of CTB is responsible for the change in permeability, we have created and tested another mutant (Δ *neuC1*) in a different gene, causing the mutant to also be incapable of binding CTB due to an independent alteration of the LOS ie inability to generate sialic acid. We decided to take the approach of creating another mutant in place of a complement because historically, complementation in *Campylobacter* is known to yield ambiguous data as complementation does not typically bring back full function in *C. jejuni*.

For example, in our *C. jejuni* fucose paper (Stahl et al, PNAS, 2011), we complemented two mutants involved in fucose biosynthesis and wrote in the discussion, "Complementation of the *cj0486* and *cj0487* strains did not restore growth, despite the partial restoration of l-fucose uptake (Fig. 2A). Not only were the transcript levels of *cj0486* and *cj0487* in the complements at very low levels (as described above), but *cj0481* was not expressed in any of the mutants or complements (Fig. S3), indicating an absence of induction of the whole operon rather than polar mutations that would result in complete loss of 3H-l-fucose uptake as observed for *cj0486* in Fig. 2A. These results are in agreement with the tight regulation of this operon by l-fucose induction and show that the low level of fucose assimilation into the complemented *cj0486* and *cj0487* strains is insufficient for induction and enhanced growth by these complemented strains."

Similarly, in our *C. jejuni* capsule paper (McNally et al, JBC, 2007), complementation of the *cj1421* and *cj1422* transferase mutants resulted in low levels of O-methyl phosphoramidate production that could only be detected by sensitive methods of ^1H - ^{31}P HSQC high resolution magic angle spinning NMR experiments that we devised to selectively detect this modification.

For this reason, we concluded that constructing a mutant in another sugar biosynthetic pathway that also results in loss of GM1 ganglioside mimicry would provide more robust data for the importance of CTB binding to gangliosides. Thus, we constructed a *C. jejuni* mutant in the sialic acid biosynthesis pathway which prevents formation of the GM1 ganglioside mimic on the bacterial lipooligosaccharide. This mutant also shows significantly decreased membrane permeability when treated with CTB, the same result observed for the galactose transferase (*cgtB*) mutant we show in the manuscript.

For the changes in fluorescence, we consulted with a biostatistician and have presented the data more simply in relative fluorescence units. We have also changed our statistics to simply compare the amount of fluorescence with and without the toxin added for each strain by two-tailed, unpaired t-tests.

Additionally, panel B should be indicated in line 680 (before *G. mellonella*).

We have added the missing indication.

3. LTB was found to cause population shifts in the intestinal microbiota of chickens. The authors suggested that it was because the toxin inhibited bacterial growth. Since the authors have identified more GM1-mimicking bacteria from chicken guts such as *Enterococcus gallinarum*, it will be interesting to see if the toxin also have any inhibitory effects on these GM1-mimicking bacteria in culture media. This would demonstrate the broader impact of the toxins and further strengthen the conclusion of the study.

To address this comment, we performed spot assays on the isolated *Enterococcus* species and did not observe toxin clearing on the agar used. However, *Enterococcus* is a Gram-positive organism so toxin impact could be different at the membrane and may simply be preventing robust bacterial colonization resulting in removal from the GI tract, or further optimization of assay conditions is necessary for this isolate. We have therefore adjusted our speculation of the role of the toxins on the microbiota to reflect this in the Results section (line 240), and in the Discussion we emphasized that even in the engineered ganglioside-mimicking *E. coli* strain, we did not see a clearing phenotype and proposed this could be due to differences in membrane architecture (lines 324-325) and now added a comment on colonization on line 381 too.

4. The image quality for Fig. 2 (D, E, and F; clearance assay) could be improved. The ones for 11168 were especially unclear.

The spot assays displayed in Fig. 2 D, E and F were repeated and this time each plate was imaged using a GO21 camera connected to an Olympus SZX16 stereoscope. We believe that the new images are much sharper now.

5. Line 33: "CTB" and "LTB" should be defined. Line 133: it was the first use of "LTB" in the main text and should be defined

Definitions for each have been added.

REVIEWERS' COMMENTS:

Reviewer #1 (Remarks to the Author):

My comments have been fully addressed.

Reviewer #2 (Remarks to the Author):

The revision has adequately addressed my concerns. I have no further comments for the authors.

REVIEWERS' COMMENTS:

Reviewer #1 (Remarks to the Author):

My comments have been fully addressed.

Reviewer #2 (Remarks to the Author):

The revision has adequately addressed my concerns. I have no further comments for the authors.

We thank the reviewers and there are no further comments to address.